# Inductive Design Exploration Method with Active Learning for Complex Design Problems

**Sungwoo Jang [1], Hae-Jin Choi [1,*], Seung-Kyum Choi [2,*] and Jae-Sung Oh [3]**

[1]   School of Mechanical Engineering, Chung-Ang University, 84, Heukseok-ro,
     Dongjak-gu, Seoul 06974, Korea; bobblejet@cau.ac.kr
[2]   The George W. Woodruff School of Mechanical Engineering, Georgia Institute of Technology,
     Atlanta, GA 30332, USA
[3]   Department of Mechanical Systems Engineering, Chung-Ang University, 84, Heukseok-ro,
     Dongjak-gu, 06974 Seoul, Korea; ojs921@cau.ac.kr
*   Correspondence: hjchoi@cau.ac.kr (H.-J.C.); schoi@me.gatech.edu (S.-K.C.);
     Tel.: +82-2-820-5787 (H.-J.C.); +1-404-894-9218 (S.-K.C.)

**Abstract:** The design of multiscale materials and products has necessitated an inductive and robust design approach to ensure satisfying the performance goals for complex engineering problems. Inductive design exploration method is a performance-driven design approach that explores feasible design spaces while considering the effect of uncertainty that leads to performance variability. However, the existing design method suffers from high computational costs for pre-defined sample data, which sacrifices the accuracy of solution spaces. In this study, we present an improved implementation of the inductive design exploration method by applying the active learning algorithm that is mainly used in machine learning techniques. The purpose of this study is to minimize the sampling effort while maintaining reasonable accuracy in the exploration of design spaces, thereby alleviating computational burden. The capabilities of the improved method are highlighted and demonstrated via a design problem of the blast resistant sandwich panel.

**Keywords:** robust design; high computational cost; active learning; complex design problem; sandwich panel

## 1. Introduction

Many engineering systems often appear with glaring features such as being hierarchical, heterogeneous, and having high-dimensionality. In addition, computationally expensive analysis/simulations make it more difficult to identify system-wide design solutions for problems. For example, a complex design problem may be decomposed into several sub-problems, each of which may have independent/dependent design variables with different goals such as disciplines and scales, among others. Typical routines of materials selection strategies are no longer conducive to ensuring system requirements while designing such complex design problems. Therefore, the nature of complex systems have necessitated the inductive multiscale design of materials and products. Olson's hierarchical concept [1] distinguished inductive and deductive decision paths in materials design. As shown in Figure 1, the deductive path corresponds to bottom–up mappings from the processing level to the performance level, whereas the inductive path is the performances-means (top–down) approach to tailor the materials structure or processing parameters to satisfy multiple performance requirements. However, there are numerous challenges due in large part to the presence of uncertainties associated with material models, design variables, and propagation across the design hierarchy. Therefore, robust designs should be considered rather than a single optimal solution to support decision-making, considering performance variation.

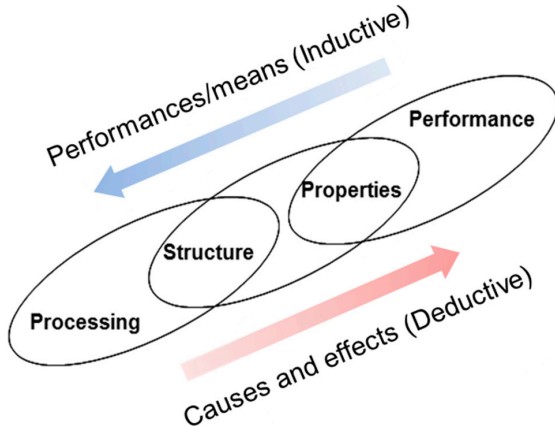

**Figure 1.** Olson's hierarchical concept of "Materials by Design" [1].

In response to these issues, set-based design approaches have become more promising as they tend to attain design requirements by exploring a set of satisfactory design solutions. Shahan et al. [2] proposed a set-based collaborative design method that classifies the design space into satisfactory and unsatisfactory regions by employing Bayesian network classifiers. This method is further applied in the hierarchical design of negative stiffness meta-materials at micro-, meso-, and macro-scales [3–5]. The inductive design exploration method (IDEM) is also a multilevel, robust design method that identifies feasible ranges of design variables in a step-by-step, top–down (inductive) manner [6–8]. In this method, feasible ranges (not an optimal point) are sequentially identified in each solution space from the top-most to the lowest level, considering the propagation of uncertainty through the model hierarchy. The detailed process in IDEM can be found in the literature for solving many complex engineering problems, including blast-resistant sandwich panels [9], vehicle-mounted antenna-supporting structures [10], ultra-high-performance concrete (UHPC) panels, etc. [11,12].

In the above-mentioned set-based approaches, one of the challenges is to tackle computationally expensive analyses and simulations. Even with the help of parallel computing, increasing computing power, reducing design space, screening significant design variables, among others, it is still difficult to deal with heavy computational costs. In IDEM, an *n*-dimensional grid of discrete points is generated at each level of a multilevel design problem; then, the grid design points are evaluated by mapping models across the design hierarchy. However, the exhaustive grid of design points can be burdensome. Meta-models are preferred for replacing computationally expensive analyses; however, they are not well-suited for models with discrete variables, nonlinear responses, or discontinuous responses, which are often found in materials design problems [5]. The prediction accuracy of meta-models is also primarily dependent on the number of experimental sample points; therefore it is difficult to build high-fidelity models with computationally expensive simulations [9]. As the number of design points to be evaluated has a direct influence on the accuracy of feasible solution spaces, designers may face difficulty in choosing an appropriate sample size to ensure reasonable accuracy. The results reported by Shahan et al. [2] also indicate the dependency of the number of initial training sample points on the shape of a feasible region boundary obtained from Bayesian network classifiers. Furthermore, it is often observed that the identified feasible regions represent a small portion of the initial design space and a large number of sample points are expended in evaluating unsatisfactory design points. Therefore, an approach is clearly needed that alleviates the exhaustive sample evaluations caused by high computational costs.

This study aims to improve the traditional IDEM to minimize the sampling effort while maintaining reasonable accuracy of feasible solution regions. To address this issue, we propose a new active sampling method motivated by the active learning technique, which is a subfield of machine learning. The specific contributions of this work are (i) feasible space representation by classification algorithm and (ii) improvement in classification boundary with an active learning

scheme. These contributions deal with the high computational expense involved in traditional IDEM. The detailed technical process is discussed in Section 2. The introduced method is then applied to the sandwich panel design as a demonstration problem and compared with the results obtained by traditional IDEM is in Section 3. The final section discusses the potential benefits of this study for further multilevel materials and product designs.

## 2. Inductive Design Exploration Method with Active Learning

### 2.1. Limitation of Traditional IDEM

The IDEM process initially proposed by Choi et al. [7,8] is briefly explained as follows. In Step 1, a rough design space at each design level is defined and it generates discrete grid points within the spaces. These discrete points are then evaluated using the employed mapping models (e.g., finite element model and surrogate model) in Step 2. This mapping or projection step is performed to relate the input design space (low level) to the output design space (high level) and repeated from the lowest level to the top-most level. Uncertainty associated with design parameters and models is taken into account in this step; thus, the evaluation of a discrete point produces a range of outputs. The evaluated discrete points with the corresponding output ranges are stored so as to help identify the feasible regions in the next step. In Step 3, feasible regions are identified in the solution spaces with given (or identified from a higher level) output ranges that satisfy the requirements. The feasible regions comprise feasible points whose evaluated output ranges are located within the constraint boundary of the requirements. These feasible points are determined based on their hyper-dimensional error margin indices (HD-EMIs). The detailed calculation of HD-EMIs is explained in Refs. [7–10]. An HD-EMI of a discrete point indicates the quantitative closeness of its output range to the nearest feasible region boundary, i.e., the robustness of a design point so as to not violate the performance constraints under its variability. An HD-EMI greater than unity for a discrete point implies that its output range lies within the constraint boundary of the performance requirements, therefore making it a feasible point. In contrast, an HD-EMI less than unity for a discrete point indicates that its output range is located outside the nearest constraint boundary, which makes it an infeasible point. With increasing HD-EMIs, the output range moves further away from the constraint boundary, thereby making it more robust.

When implementing the solution strategy of IDEM, there is a difficult question to be asked in Step 1. How many samples should we generate to identify feasible regions with acceptable accuracy? As mentioned, the number of design points to be evaluated has a direct influence on the accuracy of feasible solution spaces. Many existing studies, which utilized the traditional IDEM [6–12], also appear to show that the sample size is predetermined owing to computational cost. However, it is very difficult to choose the appropriate sample size beforehand and this difficulty is compounded when high cost computational models are involved. To deal with this issue, we present a new method using the active learning technique, which is discussed in Section 2.2.

### 2.2. IDEM with Active Learning

Active learning is a subcategory of machine learning. The key idea behind active learning is that a learning algorithm can achieve greater accuracy with fewer training data [13]. The algorithm is allowed to choose the candidate training data, specifically, the more useful data to choose, from which it learns. For any supervised machine learning to perform well, it must often be trained with abounding given labeled instances (e.g., yes or no, success or failure, feasible or infeasible, and so on). The models also can be further updated if more labeled samples are provided. However, there are situations in which unlabeled data is abundant but manually labelling is very difficult, time-consuming, or expensive. Sometimes, the labeled data may be uninformative and the trained model may thus not be useful. In such scenarios, active learning algorithms can be employed to iteratively query which samples need

to be labeled based upon past responses. As the learner chooses the samples, the sample size to learn a model can often be much smaller than that required in normal (passive) supervised learnings [13].

Figure 2 shows the general steps in active learning algorithms. The model for a certain task is first trained based on initially labeled data. The model has access to a pool of unlabeled data and can evaluate their informativeness. Given this informativeness, we ask the next query for which informativeness is highest. In other words, the most informative unlabeled data is selected as the new training data. There are several algorithms for determining which data should be labeled, i.e., query strategies, such as uncertainty sampling, query by committee, expected model change, expected error reduction, among others [13]. The selected unlabeled data is then labeled and added in the next training data set. Finally, the current model is retrained using the new training data. The steps within the for-loop are repeated until the allowable budget is reached.

---

**Train** model $M = f(initial\ training\ data)$

For $i = 1: budget$

     **Evaluate** informativeness of unlabeled data

     **Ask** query $q$ for which informativeness is the highest $\Rightarrow$ <u>Query strategies</u>

     **Add** new samples in the training data

     **Re-train** model $M$ using the new training data

End

**Return** model $M$

---

**Figure 2.** General active learning procedure.

The abovementioned active learning process is applied to traditional IDEM. The graphical procedure of the proposed method is shown in Figure 3. First, the design spaces at each level are defined, and they generate a large pool of unlabeled data. Among the unlabeled data, the initial training points to be evaluated are randomly chosen. Step 1 involves the same preparation procedure for active learning contexts. In Step 2, the initial points are evaluated using the employed mapping models to label them as feasible or infeasible. The feasibility check is performed in the same manner as traditional IDEM (same as Step 2 and 3 in Section 2.1) by calculating HD-EMIs.

The next step is to train a model with previously evaluated feasible/infeasible points and iteratively update the model in an active learning manner. Here, our model ($M$) is a classification model to represent the feasible/infeasible region in design spaces. In traditional IDEM, feasible and infeasible regions are only a group of feasible and infeasible points, respectively, and this representation is directly influenced by the initially sampled grid points at each design level. That is, a large sample size is required for the high resolution of feasible/infeasible regions, which in turn increases computational expense. Therefore, the classification algorithm is employed to avoid excessive grid points evaluations. In this study, a support vector machine (SVM) is used to classify regions as feasible or infeasible for the given feasible/infeasible samples. SVMs are one of the most common and powerful classifiers and were observed to be highly suitable for the active learning setup [14–17]. When the first SVM is used to classify feasible/infeasible regions, the initial feasible regions identified by the SVM are not expected to be accurate enough, as the number of initial training points is relatively small. Therefore, the current classifier requires more samples to update and additional sampling can be implemented through query strategies. Among the strategies, uncertainty sampling is the simplest and most commonly used query strategy that an active learner queries the instances about, which it is least certain how to label [13]. Here, the most uncertain points to be classified as feasible or infeasible are obviously the points closest to the decision boundary of the SVM. Maximum uncertainty implies that the current SVM has the least confidence on its classification. The feasibility check is performed on the chosen points through

uncertainty sampling. The newly identified feasible and infeasible points are added to the initial training points and the augmented training points are used to retrain the SVM. This closed-loop of query and retraining is performed until the allowable budget is reached or the stopping criterion is met. Finally, the final SVM classifier is used to identify feasible regions. Steps 1–3 are repeatedly performed to sequentially search the feasible solution regions at lower levels. The detailed tasks of the method are also shown in Figure 4.

■ STEP 1: Define design space and generate initial training points

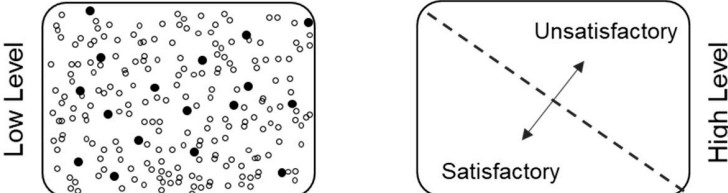

■ STEP 2: Evaluate training points using mapping functions (feasibility check)

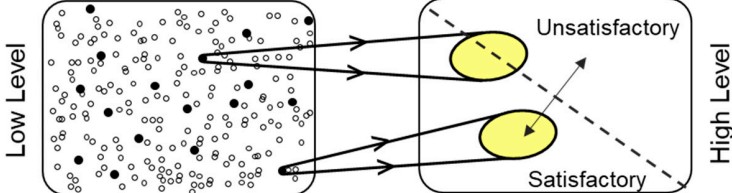

■ STEP 3: Train a classifier and update via active learning

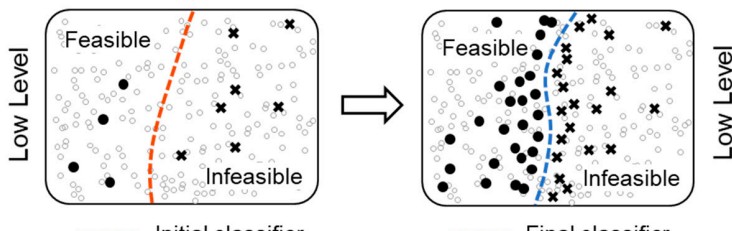

**Figure 3.** Graphical procedure of inductive design exploration method(IDEM) with active learning.

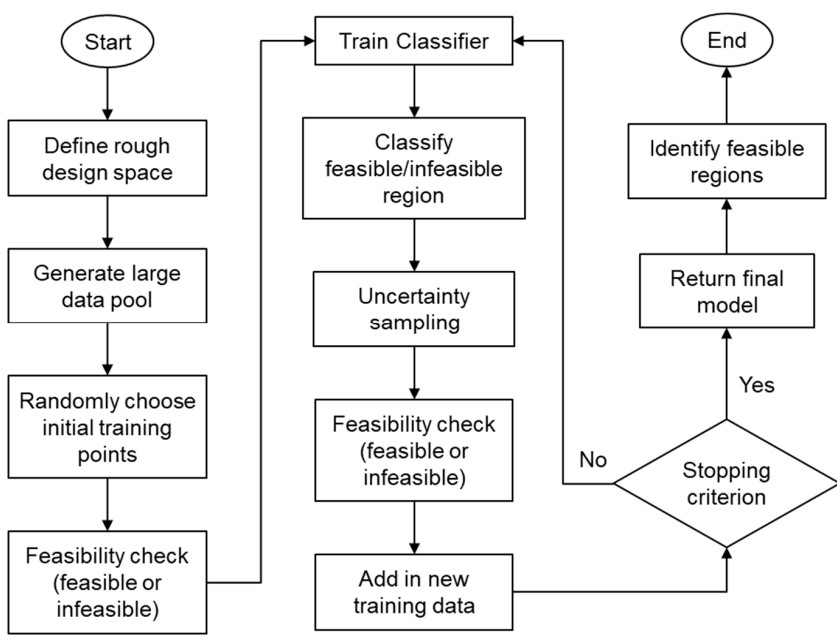

**Figure 4.** Flowchart of IDEM with active learning.

It is expected that the classification model will achieve greater accuracy with fewer total training data than those required in traditional IDEM. This is owing to the goal of active learning that selectively adds training samples to maximize accuracy with less training data. To that end, the presented IDEM with active learning is applied to the sandwich panel design as a demonstration problem and it is compared with the results obtained by traditional IDEM in Section 3.

## 3. Demonstration Problem: Design of Aluminum Foam-Cored Sandwich Panel

### 3.1. Problem Formulation

For demonstration, we designed a sandwich panel with an aluminum foam core for blast resistance applications, as illustrated in Figure 5. Aluminum foam is a useful material that is capable of absorbing blast energy or projectile impacts because of its high specific strength, low relative density, and high energy-absorbing capacity [18–21]. Given the excellent material properties, aluminum foam-cored sandwich panels (AFSPs) can be widely applied as energy absorbers in a wide range of impact- or blast-protection applications, such as crash barriers, blast mitigators, and munitions packaging. In this study, we designed a circular AFSP (2 m diameter, 100 mm fixed total thickness) capable of satisfying the performance requirements for deflection and structural mass subjected to a blast impact of 15 kg of TNT at a distance of 1 m (Figure 5).

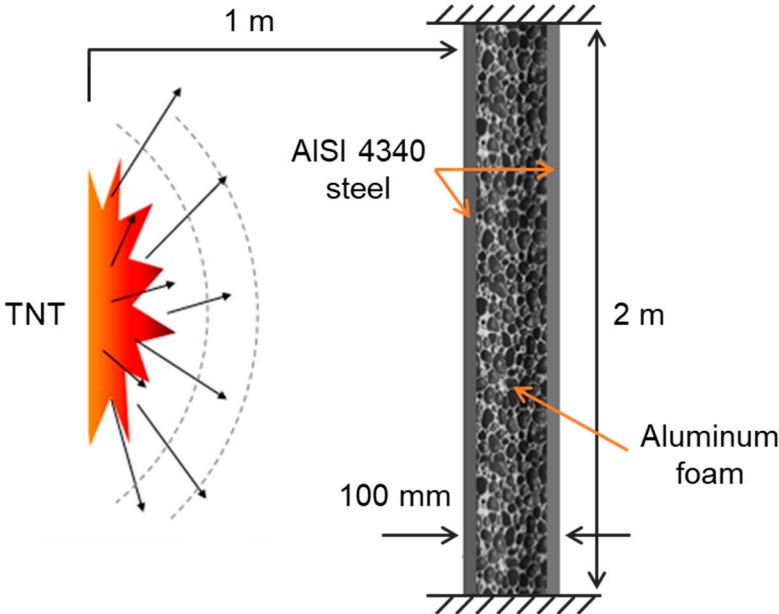

**Figure 5.** Aluminum foam-cored sandwich panels subjected to blast impact [9].

The hierarchical design flow of the sandwich panel is shown in Figure 6. The sandwich panel problem can be decoupled into three levels: Processing level, property level, and performance level. The variables designed here were the processing parameters for the aluminum foam that affect the structural/mechanical properties of the core as well as the foam core thickness ($T_c$). With the given sandwich panel geometry and material properties, the final performances were maximum deflection due to blast impact and mass/area of the panel. A response surface model links the processing and property levels and is established to represent the influence of two processing parameters (amount of $TiH_2$ and holding time) on the relative density of the aluminum foam [22]. The deflection and mass/area as functions of core density and thickness were calculated by a kriging model and mass/area equation, respectively. Additional information on the models that linked the design levels as well as model validations are discussed in detail in our previous study [9]. The performance requirements were established as a maximum deflection of ≤80 mm and mass/area value of ≤20 g/cm$^2$.

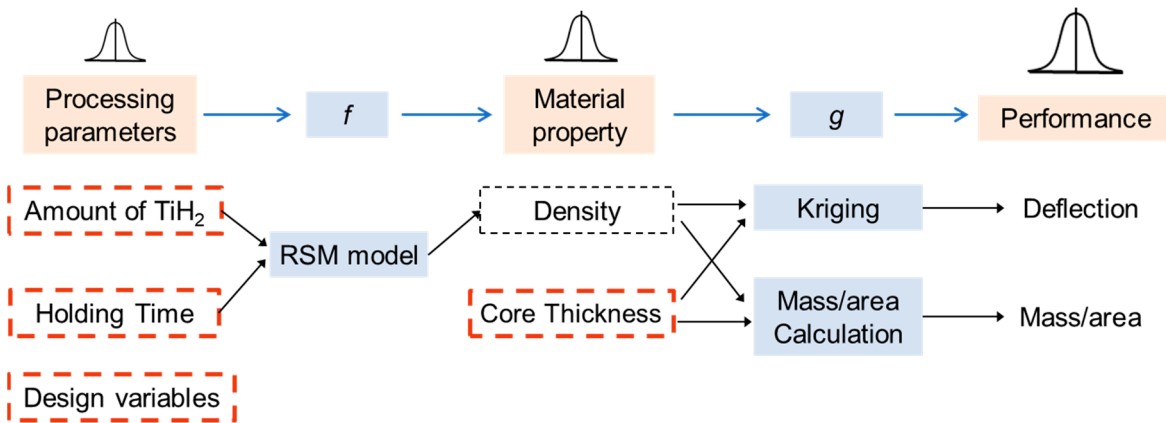

**Figure 6.** Hierarchical design flow of sandwich panel. RSM: response surface model.

### 3.2. IDEM with Active Learning for Sandwich Panel Problem

The method introduced in Section 2.2 was applied to the sandwich panel problem as following steps. The following steps of the IDEM with active learning were coded and implemented in Matlab (MATLAB Release 2018b, The MathWorks, Inc., Natick, MA., USA).

Step 1: Defined the rough design spaces for the processing parameters (holding time and amount of $TiH_2$) and geometry-property space (foam density and core thickness). A large pool of unlabeled sample data containing 2000 random samples was generated, and a total of 50 samples among these data were randomly chosen for the initial training data.

Step 2: Evaluated the chosen initial points based on the mapping models. In this case, the kriging model and mass calculation model were used as mapping models between the density—core thickness space and the deflection—mass/area space. Each sample point produces a range of outputs with an assumption of 5% variability. Based on the HD-EMIs of the evaluated data, feasible or infeasible points that satisfy the performance requirements were determined.

Step 3: Trained a classifier and updated it via active learning algorithm, as illustrated in Figure 7. First, an SVM was trained with the initial training data. The SVM is capable of nonlinear classification using kernel trick and the radial basis function kernel was used in this study [23]. The first trained SVM identified feasible/infeasible regions, and the group of boundary points as a decision boundary were drawn in a dashed red line. The true boundary obtained in the previous study is also shown in black line for comparison. To further update the SVM, new unlabeled samples m = 1 ( in Figure 7, shown in red square) that were closest to the boundary were chosen based on the uncertainty sampling. This implies that the points closest to the boundary will be the most informative samples to improve the classification boundary. Then, the samples were evaluated again and determined as feasible or infeasible points. Now, the SVM was retrained using 50 + m samples. The uncertainty sampling and retraining were repeated until the allowable budget was reached. For example, Figure 8 shows the results when 100 iterations were allowed for active learning of the SVM. As shown in the figure, the SVM decision boundary gradually improved and became similar to the true boundary, which indicates that the accuracy of the feasible region is improved by active learning.

The example shown in Figure 8 was given 100 budget iterations; however, this budget may be ambiguous in practical situations. It is also observed that the decision boundary of the SVM after 100 iterations needs to be improved further. To further improve our algorithm, the stopping criteria for the SVM active learning algorithm were established. Unlike the budget-based loop learning in the general active learning manner shown in Figure 2, two closed loops were performed with each stopping criterion in the modified algorithm. Since the true boundary is a group of points whose values of HD-EMIs are exactly unity, the stopping criteria were defined based on the HD-EMIs of decision boundary points. While increasing training points by uncertainty sampling, the first loop stopped when the mean HD-EMIs of the current SVM boundary points ≈1. This was to aggressively

update the general decision boundary of the SVM in the first loop. After the first loop is complete, there may be small areas that still do not coincide with the true boundary. In the second loop, a boundary point whose value of HD-EMI farthest to unity among the current SVM boundary points, i.e., $max\left(\left|1 - HDEMI_{boundary}\right|\right)$, was chosen as next sample to be evaluated. This corresponds to choosing a boundary point farthest to the true boundary. In this manner, the SVM boundary is consecutively corrected where the inconsistency between the current SVM and the true boundary is greatest. Then, the second loop ended when the standard deviation of the current SVM boundary points $\approx 0$. Therefore, the goal of the double-loop learning is to train the SVM whose decision boundary turns into the true boundary in two sequential loops.

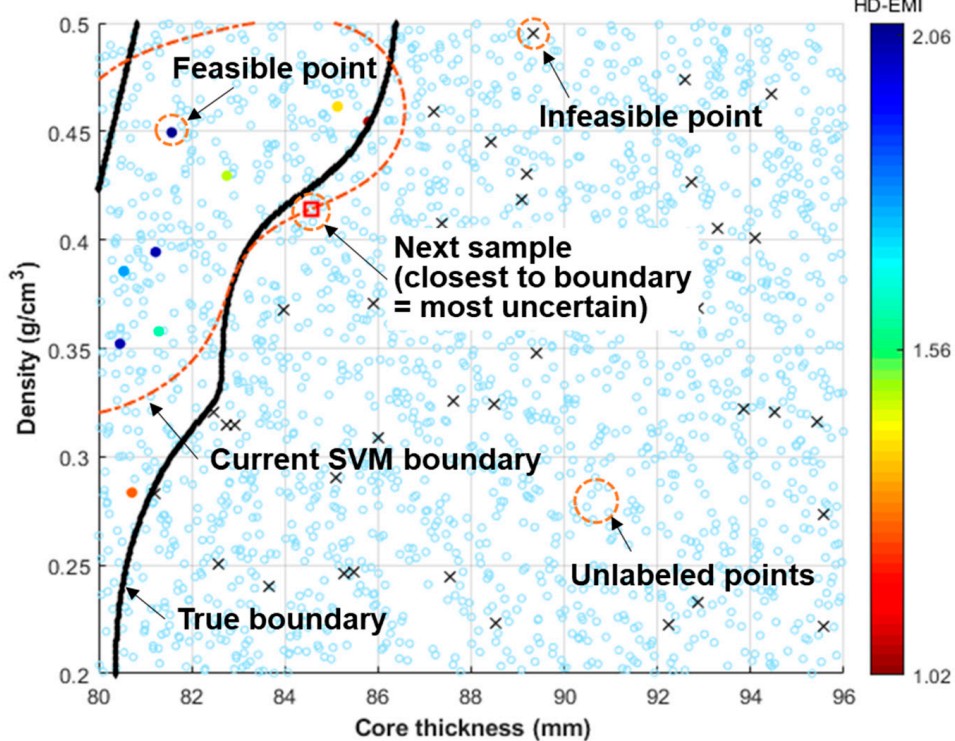

**Figure 7.** Active learning method for support vector machine(SVM).

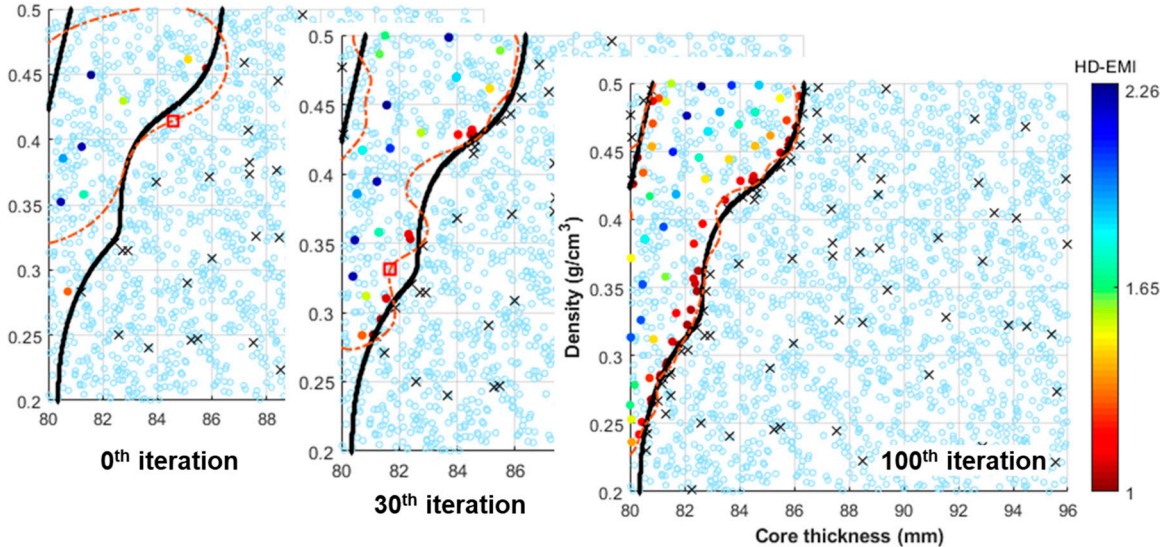

**Figure 8.** SVM active learning iterations at property level.

The modified algorithm was implemented as shown in Figure 9. In the first loop, multiple additional samples ($m = 10$) were chosen by uncertainty sampling—a batch-based sample selection at each iteration to decrease the number of times the SVM is retrained [24]. The first loop stopped after 9 iterations when $|1 - mean\_HDEMI| < 0.01$. The second loop begins to modify the small regions where the current SVM boundary is inconsistent with the true boundary. After 95 iterations, the second loop ended when $std\_HDEMI < 0.015$. The result shows that the final decision boundary of the SVM remarkably corresponded to the true boundary. The total number of samples required in the algorithm is 236 (50 initial training samples + 100 samples in 1st loop + 86 samples in 2nd loop). The average and standard deviation of the final SVM boundary were 1.007 and 0.0149 respectively, which indicates good accuracy. Therefore, it is more beneficial to utilize the double-loop active learning method since the algorithm requires no predefined budget and ends when an acceptable accuracy is achieved.

With the defined feasible region in the core density–thickness space (property level), the feasible region of processing parameters can be also found at the processing level in the same manner (Steps 1–3). Because the aluminum foam processing parameter space was related only to the foam density at property level, it was desirable to predetermine the value of the core thickness that has the largest density range to maintain as large a feasible region as possible. This way brings a designer more design freedom, thereby increasing possibility of decision making. Within the obtained feasible region of core thickness and density, the core thickness with the largest density range was selected at $T_c = 80.34$ mm. Figure 10 shows the active learning iterations used to identify the feasible region of the processing parameters that satisfied the density range at predetermined core thickness. The algorithm was implemented with the same stopping criteria and ended after 94 iterations. The total number of samples evaluated at processing level was 226 (50 initial training samples + 90 samples in 1st loop + 86 samples in 2nd loop). The result clearly shows that the feasible region was successfully obtained using the proposed active learning algorithm.

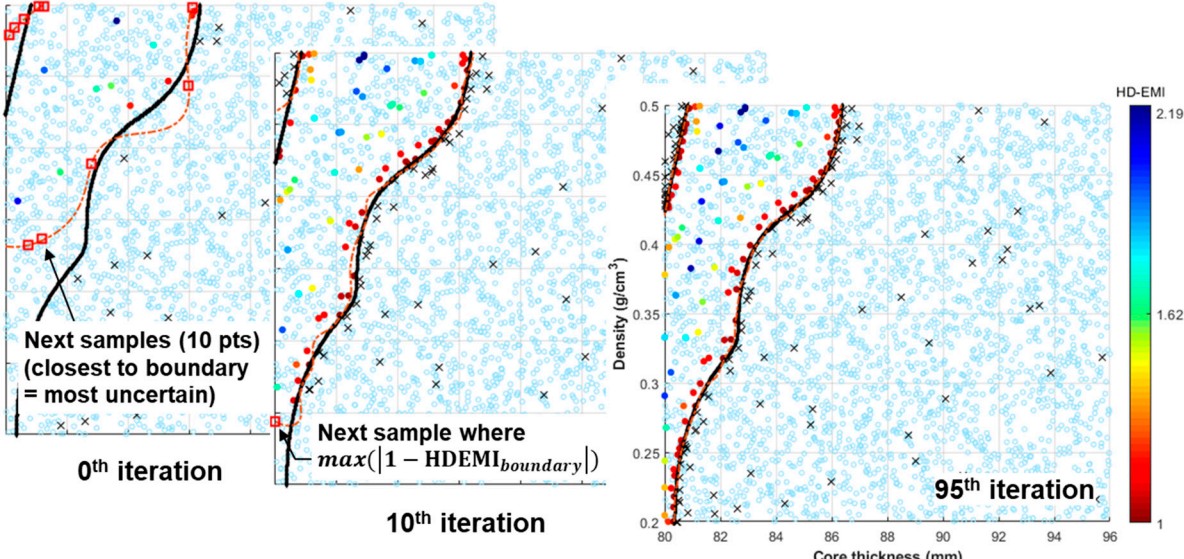

**Figure 9.** Double-loop active learning iterations at property level.

After identifying the feasible regions, the final design points can be optimally chosen within the feasible regions. The optimum design results of the sandwich panel problem as well as detailed discussion are given in our previous study [9], showing that the design solution successfully satisfies the performance requirements under uncertainty.

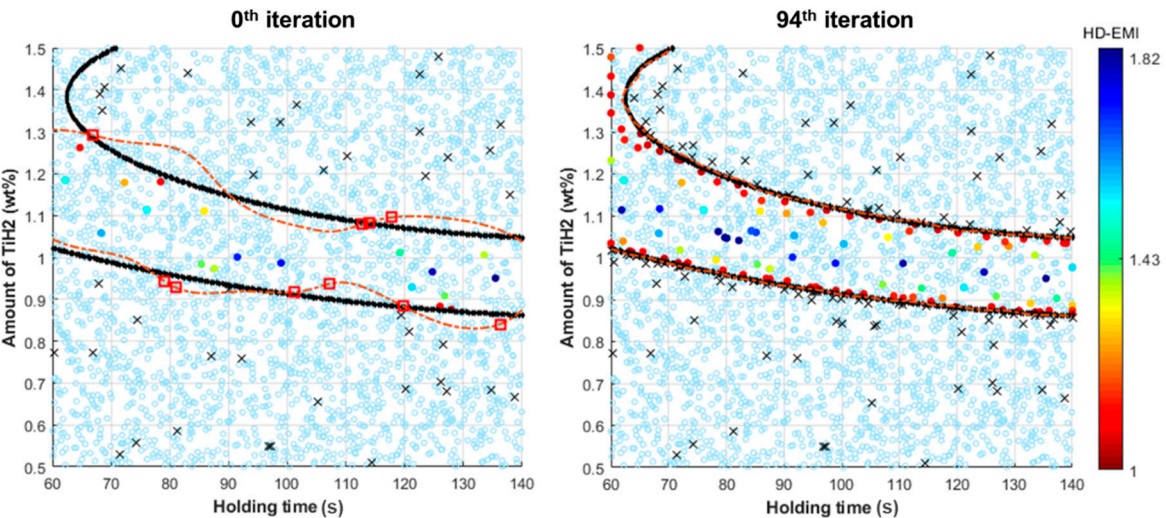

**Figure 10.** Feasible region at processing level.

### 3.3. Comparison of Results with Traditional IDEM

To discuss the usefulness of the presented active learning method, the feasible regions obtained in this study were compared to the results in traditional IDEM. In this way, the accuracy of the feasible region and the total samples required in the presented method were compared with the results in traditional IDEM. Figure 11 compares the results obtained by traditional IDEM (left) and active learning (right). The feasible region in traditional method is identified in the gridded input space (core density–thickness level) with a total sample size of 4941, that is composed of resolutions of 0.2 mm in thickness and 0.005 g/cm$^3$ in density. It can be observed that the exhaustive grid sample points were evaluated, but a large number of sample points were expended in evaluating unsatisfactory design points, which is clearly inefficient. On the other hand, the feasible region determined by SVM active learning algorithm was obtained with only 236 samples, which is much fewer yet almost identical to the true boundary.

Figure 12 also shows the comparison results obtained by traditional IDEM (left) and active learning (right) at processing level. The sample size of traditional IDEM was 4131 and composed of the resolutions of 1 s in holding time and 0.02 wt% in amount of TiH$_2$, whereas only 226 samples were required in active learning. Again, the accuracy of the SVM is evident with much less sampling effort.

As explained earlier, the initial training samples were chosen randomly, meaning that the initial SVM boundaries vary, and the total sample sizes for the final SVM to meet the accuracy criteria may be greater or smaller than those presented in this study. For example, after repeatedly running the algorithm 5 times with the same stopping criteria, it was found to have a relatively small variation in the total sample sizes (295, 175, 229, 207, and 239, respectively), which indicates the algorithm performs in a reasonably robust fashion.

In traditional IDEM, following the evaluation of the discretized input space, a numerical root finding method is used to determine the exact border of the feasible region boundary located between pairs of feasible and infeasible points. This representation of feasible regions is solely dependent on the gridded input space determined by discretizing resolution. A coarse discretization can cause a rough estimate of the feasible region composed of relatively fewer boundary points leading to some input points that are considered as satisfying but may be unsatisfactory if much finer discretization is undertaken. Furthermore, it is even more difficult in the case of arbitrary or multiple isolated shapes of the true boundary. This discretization error can be alleviated by increasing the resolution of discretization, which will, however, increase the amount of computation required.

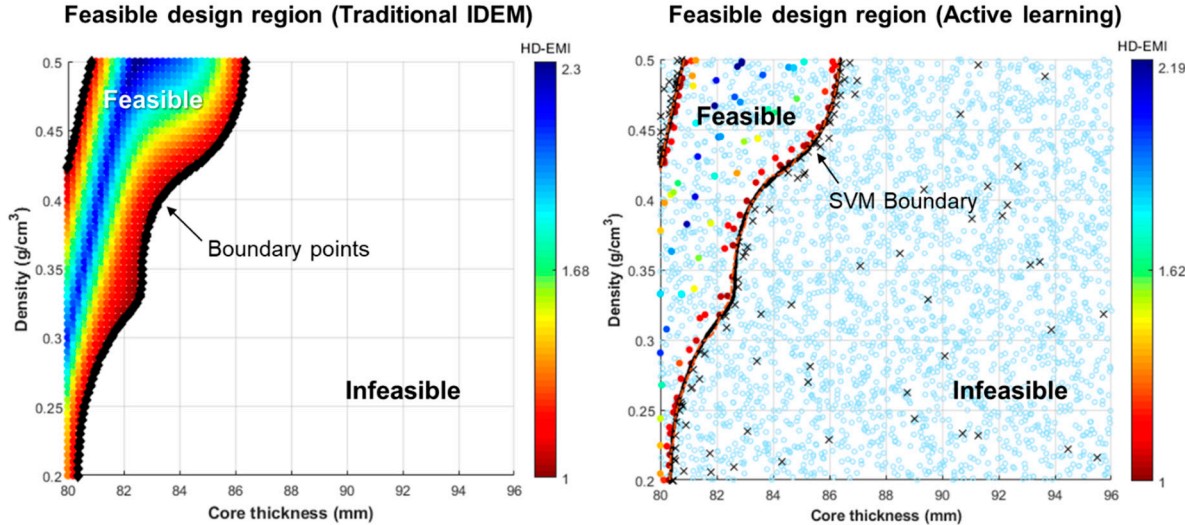

**Figure 11.** Feasible region obtained by traditional IDEM (left) and active learning (right) at property level.

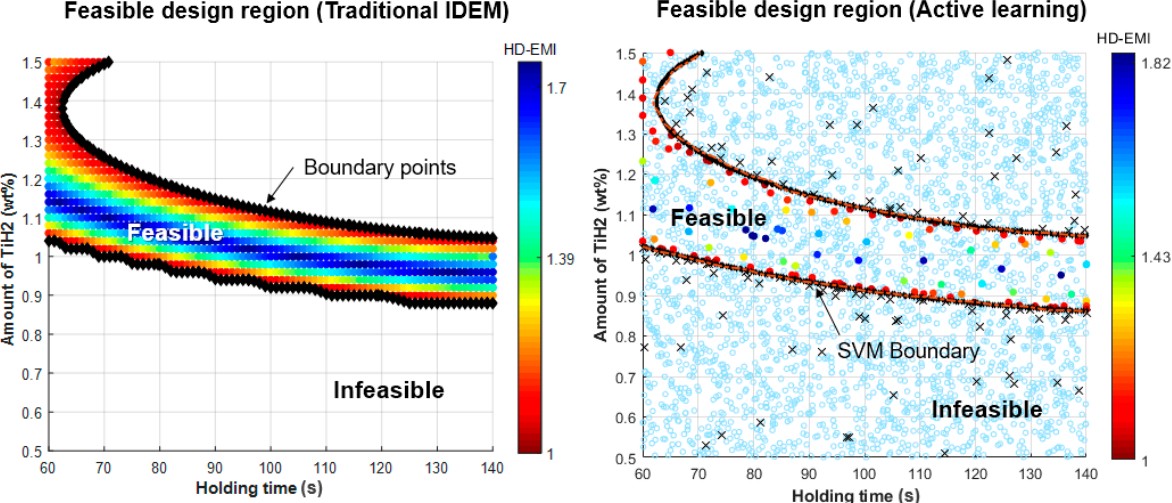

**Figure 12.** Feasible region obtained by traditional IDEM (left) and active learning (right) at processing level.

In this study, the deflection caused by the blast impact was evaluated using a kriging surrogate model; however, there may be a situation where surrogate models cannot be accepted. For example, when assuming the total 4941 samples and performing the blast impact simulations with the approximate running time of 30 min each, the total time consumed for simulations was roughly 2470 h ≈ 103 d with traditional IDEM, which is clearly time-consuming. Meanwhile, the total time for the presented method would take 118 h ≈ 5 d, which is time-saving and plausible. Overall, the comparison results indicate that the presented IDEM with active learning can significantly reduce the sample size required for identification of a feasible region.

## 4. Conclusions

This work presents an improved implementation of the inductive design exploration method (IDEM) for exploring robust solutions that can be adopted to materials and product designs. Owing to the limitations of the existing methods, the algorithm was modified so that the representation of the feasible solution region is not restricted by predefined sample size, which is uncertain in practical situations. The active learning method for support vector machine (SVM) classifier was used to determine the boundary between feasible and infeasible neighbors in a design space. The three-step process for IDEM with active learning was outlined: (i) define design space and generate initial training

points, (ii) feasibility check of sampled data, and (iii) active learning for SVM classifier. Finally, the utility of this method was demonstrated via the sandwich panel problem.

The distinct contribution of this work is that the presented IDEM with active learning not only significantly eases the sampling effort but guarantees acceptable accuracy. By reducing the number of samples, designers can quickly identify the feasible design regions where a final design selection can be made. This method will be more advantageous for complex design problems in the presence of computationally expensive simulations through the multiscale model chain.

**Author Contributions:** Conceptualization, H.-J.C.; Methodology, S.J.; Visualization, J.-S.O.; Writing—original draft, S.J.; Writing—review & editing, H.-J.C.; Writing—review & editing, S.-K.C.

**Funding:** This research was supported by the Chung-Ang University Research Grants in 2016. This research was in part supported by the Korea Institute for Advancement of Technology (KIAT) grant funded by the Korea Government - Ministry of Trade Industry and Energy (MOTIE). (No. N0001075).

**Conflicts of Interest:** The authors declare no conflict of interest.

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
