# Peer review of "Inductive Design Exploration Method with Active Learning for Complex Design Problems"

_applsci, doi:10.3390/app8122418_

Reviewer 1 Report

The problem under study is interesting but there are some issues with the article:

1) Comments about the motivation of the problems are not enough;

2) A strong comparison with similar results is missing;

3) More information about the code used should be added; 

4) Methods used to reduce the computational cost should be better described.

Author Response

Point 1: Comments about the motivation of the problems are not enough;

à We thank Reviewer 1 for the good comments. The motivation of the study is exhaustive sample evaluations caused by high computational models often involved in complex design problems. Also, many existing studies still appear to show the dependency of sample size on the accuracy of feasible solution space. Therefore, the presented study strongly aims to deal with this issue by proposing the use of active learning technique.

In response to your comment, we have revised the discussion on the motivation of the study in Introduction as well as Section 2.1. limitation of traditional method. 

Point 2: A strong comparison with similar results is missing;

à In our previous design study of blast resistant sandwich panel [9], the traditional IDEM was applied to identify the feasible design region to meet the performance requirements. In order to show the strong comparison of the same design results obtained by the traditional IDEM and the presented IDEM with active learning, Section 3.3 intensively discusses the usefulness of the new method. Figures 11 and 12 also show the direct comparison between the two results. In response to your comment, we have clearly stated the purpose of comparison results with traditional IDEM in Section 3.3, page 9, lines 277-279.

Point 3: More information about the code used should be added;

à The algorithm of newly improved IDEM with active learning is implemented in Matlab®. We have mentioned the use of Matlab software in the revised manuscript in Section 3.2, page 7, line 204.

Point 4: Methods used to reduce the computational cost should be better described.

à In order to further discuss the robustness and non-uniqueness of the IDEM with active learning, we have included the results after repeating the algorithm 5 times and shown total sample sizes required for each execution. The algorithm shows a relatively small variation in total sample sizes required for 5 executions, which indicates the algorithm performs in a robust fashion. This is discussed in the revised manuscript on page 10, lines 293-296.

Reviewer 2 Report

The paper addresses the issue of optimising the Inductive Design Exploration Method by adopting an active learning approach. The case study of an aluminium sandwich is used to validate the approach. Very well written paper easy to follow despite the very technical content. Machine learning for design is a very prominent topic. The methodology is simple but sound. It is a solid contribution that advances our understanding of machine learning for design.

Author Response

Point 1: The paper addresses the issue of optimising the Inductive Design Exploration Method by adopting an active learning approach. The case study of an aluminium sandwich is used to validate the approach. Very well written paper easy to follow despite the very technical content. Machine learning for design is a very prominent topic. The methodology is simple but sound. It is a solid contribution that advances our understanding of machine learning for design.

à We thank Reviewer 2 for very thoughtful comments. In order to further strengthen our manuscript, we have made improvements as follows.

-          Overall grammar and typing error throughout the manuscript are reexamined.

-          Consistency in terminology is maintained to avoid any confusion for readers.

-          Figure 3 is modified to clearly show the steps in IDEM with active learning.

-          The motivation of the study is further discussed in Introduction.

-          Further discussion on the robustness of the IDEM with active learning is included in Section 3.3.

Thank you very much.

Reviewer 3 Report

The manuscript introduces an inductive design exploration method (IDEM) with active learning to identify feasible and infeasible domains within hierarchical, non-linear problems.  Compared to previous IDEM algorithms, the described IDEM with active learning algorithm requires fewer sample data to identify the true boundary between feasible and infeasible domains.  As stated by the authors, such an active-learning IDEM algorithm is especially useful for concurrent materials and structure design problems.

The manuscript is well written, well reasoned, and is significant to the field of computational materials and structure design.  My comments below are meant to strengthen the manuscript for wider acceptance and use once published.  I recommend to accept the manuscript after a minor revision to address the comments below.

Major comments:

1. Page 5, Figure 3: Figure 3 is unclear in the use of "Low Level" and "High Level."  In particular:

1a. "STEP 1" appears to represent the low level (aka input) and high level (aka output) for a two-variable (e.g., x1 and x2) input space that projects to a two-variable (e.g., y1 and y2) output space.   This appearance is further emphasized by the projection of uncertainty from the input to the output shown in "STEP 2."

1b. However, the randomly selected unlabeled data points (i.e., white circles with black outlines) in the low and high levels of "STEP 1" appear to be the same except for a few black-filled circles representing training points.  Thus, Figure 3 implies that unlabeled data must be consistent at the low and high levels.  Is this correct?

1c. Placement of the right-most image "STEP 3" is confusing.  Whereas "STEP 1" and "STEP 2" show the "High Level" in the right-most image, "STEP 3" appears to show the "Low Level" with a refined SVM.  Is this correct?

2. Is it correct that "STEP 2" in IDEM with active learning incorporates "STEP 2" and "STEP 3" of IDEM (as explained in Section 2.1 on Page 4, lines 144-146).  If so, Section 2.1 might be strengthened by clearly stating so.

3. Further discussion of the robustness and non-uniqueness of IDEM with active learning would strengthen the manuscript.  E.g., Page 10-11, lines 289-290: Please include: (1) the number of times the IDEM with active learning algorithm was ran; and (2) the number of samples required for each instantiated boundary to converge to mean HDEMI to be within 0.01 and the standard deviation of HDEMI to be within 0.015.

4. Page 8, line 240-242: It is unclear how boundary points farthest from the true boundary are chosen.  Whereas the mathematical statement max( |1 - HDEMIboundary |) is clear, it is unclear what are the boundary points.  In line 244, the "true boundary" is defined as the group of points whose HD-EMIs are unity; however, I assumed that "boundary" points were different from "true boundary" points as "true boundary" points yield a trivial answer.  Please clarify the definition of "boundary" and how "boundary" points are selected.

5. Although this manuscript demonstrates that IDEM with active learning identifies a boundary more efficiently than IDEM, does IDEM with active learning require additional computations to identify optimum solutions?  Since, IDEM with active learning samples fewer points within the feasible domain than IDEM, does the feasible domain need to be tested via a regularly-spaced grid in order to find the optimum design point (cf. Page 9, lines 269-272)?

Minor comments:

1. Grammar should be reviewed throughout the manuscript.  Grammatical errors include:

1a. Subject-verb agreement on page 1, line 11: "Multiscale materials and product design has..."  The subject is plural (i.e., "Multiscale materials and product design"), but the verb is singular (i.e., "has"). 

1b. Awkward grammar on page 1, lines 15-17: "An inductive design exploration method is one of the performance-driven design approaches by exploring feasible..."

1c. Page 1, line 23: Should "resistance" be "resistant"?

1d. Page 4, line 137: Extra period before "[13]."

1e. Page 6, line 193: "pernce" undefined

2. Page 3, lines 126-128: Recommend including a reference for "As the learner chooses the samples, the sample size to learn a model can often be much smaller than that required in normal (passive) supervised learnings."

3. Page 6, lines 193-194: The performance preferences are unclear.  Do you seek a maximum deflection and maximum mass/area of the panel?  Or a maximum deflection and any mass/area of the panel?  Or, a maximum deflection and some other extremum for the mass/area of the panel?

4. Page 7, line 208: At what level(s) (e.g., holding time-amount of TiH2 space, foam density-core thickness space, both, neither, or another space) were "a total of 50 samples among these data" randomly chosen?

5. Page 7, line 213: How was "5% variability" incorporated?  More specifically, were nominal values of the input variables varied by 5%?  Or, was the 5% variability incorporated within the functions relating input to output space?  Or, something else?

6. Consistency in terminology will help readers.  In particular, Page 8, lines 229-242 refer to two "loops," whereas Page 9, line 244 states two "stages."

7. Page 8: The manuscript shows Figure 8 before Figure 7.  Figure 7 and Figure 8 should swap positions.

8. Page 9, lines 261-263: Although I agree that the largest, feasible core density range occurs at Tc = 80.34 mm, is this an appropriate design decision given that the performance preferences are undefined (cf. comment 3 above regarding Page 6, lines 193-194)? 

Author Response

The manuscript introduces an inductive design exploration method (IDEM) with active learning to identify feasible and infeasible domains within hierarchical, non-linear problems.  Compared to previous IDEM algorithms, the described IDEM with active learning algorithm requires fewer sample data to identify the true boundary between feasible and infeasible domains.  As stated by the authors, such an active-learning IDEM algorithm is especially useful for concurrent materials and structure design problems.

The manuscript is well written, well reasoned, and is significant to the field of computational materials and structure design.  My comments below are meant to strengthen the manuscript for wider acceptance and use once published.  I recommend to accept the manuscript after a minor revision to address the comments below.

Major comments:

1. Page 5, Figure 3: Figure 3 is unclear in the use of "Low Level" and "High Level."  In particular:

1a. "STEP 1" appears to represent the low level (aka input) and high level (aka output) for a two-variable (e.g., x1 and x2) input space that projects to a two-variable (e.g., y1 and y2) output space.   This appearance is further emphasized by the projection of uncertainty from the input to the output shown in "STEP 2."

1b. However, the randomly selected unlabeled data points (i.e., white circles with black outlines) in the low and high levels of "STEP 1" appear to be the same except for a few black-filled circles representing training points.  Thus, Figure 3 implies that unlabeled data must be consistent at the low and high levels.  Is this correct?

à We thank Reviewer 3 for the good comments. The unlabeled data shown in the figure are randomly generated in each space therefore, they are not consistent at both level. The low and high levels were meant to show an example of design hierarchy. However, it would be much clearer to show the low level as design space and high level as performance space for the sake of simplicity, therefore we have modified the figure of high level. The modified high level shows the performance requirements (satisfactory and unsatisfactory regions) and STEP 1&2&3 are only performed at low level (design space). The Figure 3 is modified on page 5. We are sorry for any confusion caused.

1c. Placement of the right-most image "STEP 3" is confusing.  Whereas "STEP 1" and "STEP 2" show the "High Level" in the right-most image, "STEP 3" appears to show the "Low Level" with a refined SVM.  Is this correct?

à That is correct. “STEP 3” shows only the low levels (design space) for updating SVM at initial iteration (left) and final iteration (right). There was a missing label to indicate the low level and it is corrected in the revised figure.

2. Is it correct that "STEP 2" in IDEM with active learning incorporates "STEP 2" and "STEP 3" of IDEM (as explained in Section 2.1 on Page 4, lines 144-146).  If so, Section 2.1 might be strengthened by clearly stating so.

à That is correct. The IDEM with active learning incorporates “STEP 2” and “STEP 3” of tradition IDEM explained in Section 2.1. In order to avoid any confusion, we stated this clearly in Section 2.2 on page 4, line 147.

3. Further discussion of the robustness and non-uniqueness of IDEM with active learning would strengthen the manuscript. E.g., Page 10-11, lines 289-290: Please include: (1) the number of times the IDEM with active learning algorithm was ran; and (2) the number of samples required for each instantiated boundary to converge to mean HDEMI to be within 0.01 and the standard deviation of HDEMI to be within 0.015.

à In response to your comment, the results after repeating the algorithm 5 times are included in the revised manuscript on page 10, lines 293-296.

4. Page 8, line 240-242: It is unclear how boundary points farthest from the true boundary are chosen.  Whereas the mathematical statement max( |1 - HDEMIboundary |) is clear, it is unclear what are the boundary points.  In line 244, the "true boundary" is defined as the group of points whose HD-EMIs are unity; however, I assumed that "boundary" points were different from "true boundary" points as "true boundary" points yield a trivial answer.  Please clarify the definition of "boundary" and how "boundary" points are selected.

à The SVM shown in red dashed line in Figure 7 is also a group of decision boundary points. We have clarified that the SVM decision boundary is a group of boundary points in Section 3.2, line 217. The mathematical statement,                                                indicates the point whose value of HD-EMI is the farthest to unity among the current SVM boundary points, i.e., a point farthest from the true boundary points whose values of HD-EMI equal to 1. The true boundary shown in the figure is only used for the comparison. To avoid any confusion, we have clarified this step on page 8, lines 236-241.

5. Although this manuscript demonstrates that IDEM with active learning identifies a boundary more efficiently than IDEM, does IDEM with active learning require additional computations to identify optimum solutions?  Since, IDEM with active learning samples fewer points within the feasible domain than IDEM, does the feasible domain need to be tested via a regularly-spaced grid in order to find the optimum design point (cf. Page 9, lines 269-272)?

à The final design can be chosen “within” the identified feasible region (not “among the feasible points”), therefore there is no need for additional grid samples within the feasible region. The sentence was miswritten, and this is fixed in the revised manuscript, page 9, lines 271-272. The optimum design point in our previous study [9] was chosen by maximizing HD-EMIs within the feasible space. We are sorry for the mistake.

Minor comments:

1. Grammar should be reviewed throughout the manuscript.  Grammatical errors include:

1a. Subject-verb agreement on page 1, line 11: "Multiscale materials and product design has..."  The subject is plural (i.e., "Multiscale materials and product design"), but the verb is singular (i.e., "has").

à We have corrected the subject as “The design of multiscale materials and product” for the singular verb in the revised manuscript.

1b. Awkward grammar on page 1, lines 15-17: "An inductive design exploration method is one of the performance-driven design approaches by exploring feasible..."

à We have corrected the sentence as “Inductive design exploration method is a performance-driven design approach by exploring feasible…” for clearer message.

1c. Page 1, line 23: Should "resistance" be "resistant"?

à It is corrected in the revised manuscript.

1d. Page 4, line 137: Extra period before "[13]."

à It is corrected in the revised manuscript.

1e. Page 6, line 193: "pernce" undefined

à It is corrected as “performance” in the revised manuscript.

2. Page 3, lines 126-128: Recommend including a reference for "As the learner chooses the samples, the sample size to learn a model can often be much smaller than that required in normal (passive) supervised learnings."

à A reference for the phrase is included in the revised manuscript.

3. Page 6, lines 193-194: The performance preferences are unclear.  Do you seek a maximum deflection and maximum mass/area of the panel?  Or a maximum deflection and any mass/area of the panel?  Or, a maximum deflection and some other extremum for the mass/area of the panel?

à It was not intended to specify the performance preferences in the phrase but rather, introduce the performance variables. However, the maximum deflection here indicates the maximum deflection at the back face of the deformed panel due to blast impact. We have corrected the sentence as the phrase “to be achieved” may confuse readers. The performance requirements for the maximum deflection and mass/area are given to be less than or equal to 80 mm and 20 g/cm2, respectively.

4. Page 7, line 208: At what level(s) (e.g., holding time-amount of TiH2 space, foam density-core thickness space, both, neither, or another space) were "a total of 50 samples among these data" randomly chosen?

à The 50 initial training samples are chosen at both levels since the STEP 1-3 in Section 3.2 are performed at both levels in the same manner. We have specified that the STEP 1-3 are also performed at processing parameter space on page 8, lines 259-260.

5. Page 7, line 213: How was "5% variability" incorporated?  More specifically, were nominal values of the input variables varied by 5%?  Or, was the 5% variability incorporated within the functions relating input to output space?  Or, something else?

à In this study, model uncertainty is assumed to have 5% variability for the sake of simplicity. However, the function variability can be estimated using an error propagation method if the degree of manufacturing uncertainty in each input is preliminarily known. The further discussion is included in our previous study [9].

6. Consistency in terminology will help readers.  In particular, Page 8, lines 229-242 refer to two "loops," whereas Page 9, line 244 states two "stages."

à As you pointed out, we have replaced the term “stages” with “sequential loops” for consistency in terminology on page 8, line 245.

7. Page 8: The manuscript shows Figure 8 before Figure 7.  Figure 7 and Figure 8 should swap positions.

à Figure 7 has been repositioned at the bottom of page 7 for correct figures order.

8. Page 9, lines 261-263: Although I agree that the largest, feasible core density range occurs at Tc = 80.34 mm, is this an appropriate design decision given that the performance preferences are undefined (cf. comment 3 above regarding Page 6, lines 193-194)?

à The identified feasible region indicates the multiple solution range that satisfy the abovementioned performance requirements for sandwich panel. Since the aluminum foam processing parameters are only related to the foam density at property level, the feasible region in processing parameter space can only be identified with fixed value of the core thickness. In this study, we predetermined the value of core thickness that has the largest feasible density range as an example. The reason behind this is that this way brings a designer more design freedom by maintaining feasible region as large as possible, thereby increasing possibilities of decision making. We have included more justification for this in the revised manuscript on pages 8-9, lines 262-263. Thank you very much for your thoughtful comments.

Round  2

Reviewer 1 Report

The authors have improved the presentation of their article and in my opinion it can be published in this form